# Impact of Effective Intravesical Therapies on Quality of Life in Patients with Non-Muscle Invasive Bladder Cancer: A Systematic Review

**DOI:** 10.3390/ijerph191710825

**Published:** 2022-08-30

**Authors:** John W. Yuen, Ricky W. Wu, Shirley S. Ching, Chi-Fai Ng

**Affiliations:** 1School of Nursing, The Hong Kong Polytechnic University, Hung Hom, Kowloon, Hong Kong SAR, China; 2Biological and Biomedical Sciences, Glasgow Caledonian University, Glasgow G4 OBA, UK; 3S.H. Ho Urology Centre, Department of Surgery, The Chinese University of Hong Kong, Shatin, Hong Kong SAR, China

**Keywords:** quality of life, health-related quality of life, non-muscle invasive bladder cancer, urothelial carcinoma, intravesical therapy, Bacillus Calmette–Guérin, chemotherapy, immunotherapy, hyperthermia, bladder cancer

## Abstract

Background: Conventional and newly emerged intravesical modalities have demonstrated prophylactic effectiveness that may improve quality of life (QoL) in non-muscle invasive bladder cancer. The purpose of this study is to analyze existing QoL evidence in patients receiving any form of intravesical therapy. Methods: A PubMed search without time restriction was conducted to identify all relevant studies in accordance with the PICOT question. Additionally, a search was also performed in the Cochrane library database, Internet, and citation. The CONSORT 2010 checklist and STROBE statement checklist were used to evaluate the risk of bias of the included studies. Results: A total of 24 eligible articles were included, which consisted of 11 interventional and 13 observational studies. Intravesical therapy with Bacillus Calmette–Guérin (BCG) or certain chemotherapeutic agents worsens symptom burdens and functional performance during the initial induction phase while continuous improved is observed throughout the maintenance treatment and beyond. Hyperthermia has shown a positive trend in enhancing QoL of patients receiving intravesical chemotherapy, which requires more investigations. However, QoL data were unavailable for other forms of immunotherapy, immune checkpoint inhibitors, electromotive drug administration, and photodynamic therapy. Conclusions: Limited studies suggested the long-term positive impact of intravesical BCG immunotherapy and chemotherapy. However, existing evidence was lacking to clarify the impact of many emerging intravesical therapies that have suggested to be effective and safe, which demands treatment-specific QoL studies.

## 1. Introduction

Bladder cancer ranked the 10th most common cancer with 573,000 newly diagnosed cases worldwide in 2020, which is dominated by men at approximately 4 times higher prevalence than women [1]. Non-muscle invasive bladder cancer (NMIBC) accounts for approximately 70% of new cases of bladder cancer, which consists of three early-stage diseases: papillary transition carcinoma (Ta), flat carcinoma in situ (CIS), and cancer confined to the submucosa (T1) [2]. Despite having transurethral resection of bladder tumor (TURBT) as a curative-intent treatment, the 1-year and 5-year recurrence rates remain high at 15–61% and 31–78%, respectively [3], whilst 20–25% of initially NMIBC would progress to muscle-invasive stages during the lifetime, which is indicative for radical cystectomy resulting in urinary diversion to impair patients’ quality of life (QoL) [4,5]. Adjuvant prophylactic chemo- or immunotherapy is often provided to NMIBC patients following TURBT [6]. A recent survey conducted in Brazil reported that almost 79% and 35% of urologists utilized adjuvant intravesical therapy in their patients with high-risk and low-risk bladder cancer [7]. Frequent relapses of the disease require long-term surveillance follow-up accompanied with protracted course of treatment and care that compromises patients’ QoL [8].

Many researchers [6,8,9,10,11] have reviewed QoL during the NMIBC trajectory, but existing treatment-specific QoL studies were mainly restricted to muscle-invasive progression [10,12] or cystectomy [10]. Efficacies of various intravesical agents used for NMIBC were evaluated in systematic reviews, whereas QoL outcomes were usually missing [13]. Intravesical instillation maximizes lesions exposure to the therapeutic agent at lower dosage since it does not involve absorption and systematic metabolism [14]. Despite numerous receptors expressed on the surface bladder urothelium also facilitating some of the therapeutic actions to take place, treatment efficacy is largely dependent on effective drug penetration into bladder tumors prevented by the distinctive tight junction and negatively charged glycosaminoglycan (GAG) layer [14,15]. Chemical and physical strategies such as electromotive drug administration (EMDA) are trending modalities used for improving the permeability of instilled drugs [14,15]. However, lacking data of disease-specific QoL has restricted quantitative analysis in other systematic reviews reporting individual intravesical treatment modalities, including chemotherapy [16,17], immunotherapy [18,19,20,21], and EMDA [22]. The critical review by Botteman et al. [23] summarized the findings of two prospective cohort studies and concluded that post-TURBT intravesical BCG instillation has no serious impact on life satisfaction with NMIBC patients, notwithstanding some unpleasant micturition symptoms experienced at the initial stage while these symptoms were improved into the maintenance phase. The latest systematic review [8] shared the same insight while inconclusive evidence has suggested alternative intravesical treatment for better QoL. Furthermore, two phase III effectiveness trials that reported patient experience during intravesical instillation were identified in another systematic review [24], which only summarized the QoL instruments and methods instead of the outcomes.

Even though Bacillus Calmette–Guérin (BCG) remains as an effective intravesical agent for treating NMIBC after 40 years of practice, new modalities have emerged demonstrating at least comparable prophylactic effectiveness but fewer toxicities, and circumvent the need for radical cystectomy in progressed cases that may improve patients’ QoL [25,26]. Aside from modifying the treatment schedule and dosage of current intravesical agents, current boosting strategies include sequential and combination therapy [19,20], hyperthermia [27,28,29], and EMDA [22,30] that required devices to assisted drug delivery, priming immune checkpoint blockade [31], and novel photodynamic approaches [32,33]. However, the QoL impact of these effective intravesical treatments has not been addressed. The current systematic review is conducted to gather such information.

## 2. Materials and Methods

This systematic review methodology followed the Preferred Reporting Items for Systematic Reviews and Meta-Analysis guidelines [34]. The PICOT question ‘*Are NMIBC patients (P) who have received intravesical therapy (I) compared with those without or received other treatments (C) experiencing better quality of life (O) throughout the cancer trajectory (T)?*’ was used to guide the search strategy and evidence acquisition process.

In May 2022, a PubMed search without time restriction was conducted using the formulated search terms algorithm developed from Medical Subject Headings (MeSH) to identify all relevant studies in accordance with the PICOT components:*Population (P)*: (“carcinoma, transitional cell” [MeSH Terms] OR (“urinary bladder neoplasms/drug therapy” [MeSH Terms] OR “urinary bladder neoplasms/prevention and control” [MeSH Terms] OR “urinary bladder neoplasms/therapy” [MeSH Terms]) OR “Bladder cancer” [All Fields] OR “urothelial cancer” [All Fields] OR “Non-muscle invasive bladder cancer” [All Fields] OR “NMIBC” [All Fields] OR “urinary bladder neoplasm*” [All Fields])*Intervention (I) and (C)*: (“Immunotherapy” [MeSH Terms] OR “chemotherapy, adjuvant” [MeSH Terms] OR “Drug therapy” [MeSH Terms] OR “intravesical therap*” [All Fields] OR “immunotherap*” [All Fields] OR “chemotherap*” [All Fields] OR “drug therap*” [All Fields] OR “electromotive drug administration” [All Fields] OR “EMDA” [All Fields] OR “Photochemotherapy” [MeSH Terms] OR “Aminolevulinic Acid” [MeSH Terms] OR “5-ALA” [All Fields] OR “ALA” [All Fields] OR “Photodynamic” [All Fields] OR “PDT” [All Fields] OR “PDD” [All Fields] OR “thermochemother*” [All Fields] OR “Hyperthermia” [MeSH Terms] OR “hyperthermia, induced” [MeSH Terms] OR “hyperther*” [All Fields])*Outcome (O)*: (“Quality of life” [Title/Abstract] OR “Quality of life” [MeSH Terms]”)

Search terms for (I) and (C) included all existing intravesical therapeutic approaches and were identical because studies may compare between different forms of intravesical therapies. No search terms for the time (T) component were created for maximizing the outcome to cover the whole caner trajectory period. The Boolean operator “AND” was used to combine the sets of search terms between PICOT components. The Cochrane library database, Internet, and citation searching was also conducted.

Two independent reviewers (J.W.Y. and R.W.W.) carried out the two-step process: (1) title/abstract screening and (2) further full-text assessment following duplication removal, according to the eligibility criteria. Any disagreements were resolved through discussion among the reviewers and team consensus. The same procedure also applied to the quality assessment and data extraction of selected studies. The eligibility criteria were:Original research articles reporting full or part of interventional and observational studies (e.g., pilot clinical trial, baseline profile of cohort study) encompassed at least one arm of intravesical therapy in analysis,Involved NMIBC patients at stage of T0-1 and CIS of any tumor grades,Operationization of main QoL variables, andStudies in English language.

In contrast, studies reported a mixed population of NMIBC and advanced stage patients were excluded.

### 2.1. Assessment of Quality and Risk of Bias

Quality of the selected studies was assessed using the CONSORT 2010 checklist for interventional studies and the STROBE statement checklist for observational studies. Risk of bias results of each individual study were charted.

### 2.2. Data Extraction and Synthesis

One reviewer (J.W.Y.) extracted data, which was cross-checked by a second reviewer (S.S.C.). The following information was recorded: first author, publication year, study design, cancer stage and grade condition, intravesical agents used, comparison groups and sample size, QoL instruments variables used, time points of measurement, and main findings. The main findings recorded outcomes of general and bladder cancer treatment-specific QoL variables, numeric (e.g., percentage, mean scores, standard deviations) or categorical (e.g., Likert, satisfaction level) expressions, significance levels, changes, and trends among study groups and time points of measurement. Relevant information was also extracted from the supplementary documents of individual studies.

## 3. Results

### 3.1. Search Results and Study Selection

#### 3.1.1. Characteristics of Included Studies

As summarized in the PRISMA flow diagram (Figure 1), the literature search conducted in May 2022 identified 585 records following removal of duplicates. Among the 91 full-text records retrieved and assessed against the eligibility criteria, 66 of them were excluded with 48 due to lacking QoL outcome measures, 11 for not including a study arm of intravesical therapy or NMIBC in the reports, and 8 were not original studies. All eligible studies were in quantitative research design, except for one cross-sectional that used a mixed method approach including a quantitative survey and qualitative interview [35].

Table 1 summarizes the detailed characteristics of the 24 studies. They were published from 1996 to 2022, and consist of 11 interventional and 13 observational in design. A total of 2098 patients of any NMIBC stages and grades who received any forms of intravesical therapy were involved and distributed in the 4 cross-sectional (N = 620), 9 prospective cohort (N = 762), 2 single-arm clinical trial (N = 70), and 9 randomized controlled trial (RCT) studies (N = 646). Particularly, literature by Hayne et al. [36] is a protocol article that is embedded with the pilot RCT results. Regarding the intravesical agents assessed, almost 60% of the studies involved BCG (14 out of 24) with 6 of them having a design to compare the effects with a chemotherapeutic agent. While 4 studies did not specify the intravesical agent used, which were all cross-sectional surveys, 10 studies involved chemotherapeutic agents (including cisplatin, gemcitabine, mitomycin C, and pirarubicin) with 3 of them assisted by hyperthermia and 1 in combination with hyaluronic acid. Additionally, a single study evaluated the effects of an immunotherapeutic agent, interleukin-2, on QoL.

#### 3.1.2. Risk of Bias

Overall quality of the interventions included in this systematic review was poor because the majority of these studies carried high or unclear risk of bias in at least half of the 7 evaluated components (Figure 2a). The risk for performance bias (91%) and detection bias (100%) were exceptionally high due to lacking evidence of blinding procedures (Figure 2a,b). Blinding of patients but not the outcome data was performed in the study of Bosschieter et al. [51] while Tan et al., 2019 [52] fulfilled all remaining components except blinding. The performance regarding selection bias was moderately poor, because 7 and 9 trials were rated high or unclear risk for random sequence generation and allocation concealment, respectively, including the 2 single-arm interventions [42,58] in which such procedures are completely lacked by default (Figure 2a). Risk of attrition bias by analyzing incomplete outcome data was rated as low in 82% (Figure 2a,b). In contrast, approximately one-third (36%) of the included studies carried unclear risk of reporting bias due to selective reporting of outcome results for further analysis (Figure 2a,b). However, other unclear forms of bias were identified in most of the included interventions (Figure 2a,b).

For observational studies, quality assessment results were summarized in Figure 3a and grouped into 6 areas for reporting publications in Figure 3b. Almost all included studies have adequately explained the scientific background, rationale for investigation, and specific objectives or hypothesis; however, approximately one-third (39%) of them failed in indicating the study design in the title or abstract (Figure 3a,b). Regarding the methods, none of the studies has described any efforts to address potential source of bias; in addition, 85% and 77% of the included studies inadequately explained ‘how the sample size was arrived at’ and ‘statistical approach for addressing missing data’, respectively (Figure 3a,b). When reporting the results, 77% of studies fulfilled the requirement of indicating the numbers of individuals at each stage but reasons for non-participation were not given in half of them. Participant characteristics, exposure information, or missing data per each variable of interest were not reported in 85% of studies, whereas only 54% adequately reported the main finding with unadjusted estimates and their precision and 69% did not perform other analyses such as subgroup analyses and sensitivity (Figure 3a,b). Key results and cautious overall interpretation were adequately discussed in general, but only 7 (54%) and 1 (7.7%) of the included studies addressed the study limitations and generalization of the results, respectively (Figure 3a,b). However, up to 70% of the studies did not report the source of funding and the role of the funders (Figure 3b). Overall, the cohort study conducted by Schmidt et al. [47] was rated as a high-quality report which fulfilled almost all 22 STROBE reporting items. In the remaining studies included in this review, 7–18 items were inadequately reported (Figure 3a).

### 3.2. QoL Measures and Instruments

#### 3.2.1. General QoL

Earlier research asked general QoL questions using self-constructed scales to measure satisfaction to life and health status [37], overall QoL [38], and satisfaction of general QoL and affecting daily life [35]. William-Cox [35] also measured particularly QoL related to physical condition using a 7-point scale in addition to qualitative interviews. Other researchers also measured multiple additional aspects related general QoL, which included sexual activity, symptoms, daily activity levels [38], working activity relations, sexual couple life, and self-esteem [39]. Huang et al. [46] used the Visual Analogue Scale (VAS) to measure pain intensity while Miyake et al. [57] measured sleep quality using the MotionWatch8 mobile application.

The EORTC QLQ-C30 questionnaire was identified as the most frequently adopted validated instrument for measuring the global health status [43,44,45,48,50,53,54,55,57,58]. Three other studies measured health states of participants using the EQ-5D questionnaire [52,54,55], while general health was measured in two studies using the SF-36 heath survey [47,51] and SF-8 which is the abbreviated version of an original 36-item [57]. Within the context of generic QoL, these instruments also cover functioning and symptoms of physical, mental, and social health in certain extents:EORTC QLQ-C30: Contains 30 items to measure Global health status, functioning QoL (5 subscales), symptom QoL (8 subscales), and financial difficulties [59].EQ-5D: Severity levels of 5 dimensions including mobility, self-care, usual activities, pain & discomfort, and anxiety and depression [60].SF-36: Contains 36 items to measure 8 domains including physical functioning, role limitations due to physical problems, role limitations due to emotional problems, general health, bodily pain, vitality, social functioning, and mental health [61].SF-8: Contains 8 items to measure the same 8 domains of SF-36 [62].

#### 3.2.2. Disease- and Treatment-Specific QoL

Numerous researchers measured the QoL more comprehensively using combination of a general QoL instrument with one or more other disease- or treatment-specific tools. The EORTC-BLS24 is a NMIBC-specific module of the QLQ-C30 for assessing health-related QoL that they were commonly coupled together and used in some studies [36,44,50,51,58]. Danielsson et al. [49] developed their own 17-item questionnaire to measure the urinary bladder symptoms and burdens in NMIBC patients treated with BCG instillation. Moreover, two other bladder cancer-specific instruments were identified from the studies included in this review, they are BCI [40,47] and FACT-BI [56,57]. The domains of the three bladder cancer-specific instruments are listed:EORTC QLQ-BLS24: Modular questionnaire specific for NMIBC which contains 24 items grouped into 6 scales to measure the QoL affected by urinary symptoms, malaise, intravesical treatment issues, future worries, bloating and flatulence, and sexual function. Later in 2014, it was updated to QLQ-NMIBC24 which has maintenance but added 5 sexual function subscales [63].BCI: Contains 36 items assessing QoL in urinary, bowel and sexual health domains function among patients with bladder cancer [64].FACT-BI: Contains 39 items to measure 5 domains including physical, social, emotional, functional well-being, and a bladder cancer subscale [65].

The BCI was used alone in one study [40] while it was used to supplement the general QoL measured by using the SF-36 questionnaire in the study Schmidt et al. [47]. Other researchers measured how QoL is affected by urinary symptoms and sexual functions by adopting instruments commonly used in urology other than bladder cancer, they are:IPSS: Contains 8 items to measure severity of urinary symptoms with the last item (IPSS-8) asking the QoL living with the urinary symptoms [66].IIEF-5: Contains 5 items to measure erectile function with the last item addressing intercourse satisfaction [67].ICSI: Contains 4 items measuring voiding and pain without asking any QoL aspects [68].CLSS: Contains 10 items addressing the frequency of lower urinary tract symptoms, and additional 2 single items asking the single symptom which had the greatest impact and the feeling of living with the urinary condition for the rest of life [69].

Although it was first established for addressing lower urinary tract symptom secondary to benign prostatic hyperplasia, the IPSS is frequently adopted by urologist to measure urinary symptoms and burdens of patients with different urological diseases [66]. Two studies used IPSS together with FACT-BI [56,57]. Particularly, Miyake et al. [57] also measured the general QoL using QLQ-C30 and SF-8 as well as sleep quality using the MotionWatch8 mobile application, in addition to IPSS and FACT-BI but a bladder cancer-specific QoL measurement was missing. On the other hand, the RCT conducted by Hayne et al. [36] has comprehensively evaluated the effect of a new chemotherapeutic modality by using a set of instruments including QLQ-C30, QLQ-BLS24, IPSS, and ICSI. One study used IPSS in addition to IIEF-5 which is a simplified version of the International Index of Erectile Function [70] to address in particular the effect of intravesical BCG therapy on QoL associated with urinary symptoms and erectile function [41]. Wittlinger et al. [42] only reported the results of IPSS-8 as a summary of how the patients living with urinary symptoms upon an intravesical treatment. The CLSS was identified in a single study and used in addition to QLQ-C30 to evaluate the effect of an intravesical chemotherapy on general QoL and QoL living with lower urinary tract condition in intermediate- or high-risk NMIBC patients.

### 3.3. Impact of Intravesical Therapies on QoL

#### 3.3.1. Is Intravesical Therapy Better Than Other Treatments?

Four cross-sectional studies did not specify the intravesical agents that were exposed to the surveyed patients. When compared with patients without receiving TURBT alone or other non-specified treatments, consistent findings suggested that intravesical therapy caused some degrees of burdens on sexual functions especially in men [39,40,54]. Apart from sexual/couple life, Abbona et al. [39] indicated the alteration of working activity/relations and self-esteem by intravesical therapy in a quite significant number of patients when reporting their baseline measurements before recruiting them into a psychosexual support therapy. Pain and discomfort were also more frequently reported in patients who received intravesical instillation following TURBT, but an overall better emotional quality as compared with TURBT alone [54]. When compared with surgical treatment, intravesical chemotherapy with MMC resulted in constant deteriorations in urinary symptoms and global health status throughout the 12-month follow-up period [53]. The benefit of intravesical therapy on mental health over TURBT was coherent with the findings of a cohort study [47]. In a recent RCT study [55], when compared with radical cystectomy (RC), intravesical BCG therapy was shown to have negative impacts on urinary symptoms and sexual functioning without demonstrating remarkable improvement in general QoL, especially during the initial 3–6 month treatment period. However, the negative impacts of intravesical therapy seemed to be associated with the induction period but improved after the maintenance [35], resulting in a possible better global QoL state over time along the treatment [35,54]. Such pattern of improvements observed in various symptom and functioning parameters following the induction were consistent with another cohort study [50].

#### 3.3.2. BCG Immunotherapy

Intravesical BCG instillation exhibited long term benefits in reducing emotional burdens caused by urinary and sexual symptoms as soon as at 1 month [41] and up to 5 years after completing the treatment [49]. Patients who received BCG therapy maintained stable health status in general over time [41,49]. In the RCT conducted by Catto et al. [55], as measured by the QLQ- NIMIBC24, worsening of urinary symptoms and concerns about contaminating a partner were observed in the first 3–6 months upon commencing the treatment schedule, but restored to the baseline levels with small reductions in future worry scores in the next 6 months. This is consistent with the findings indicated deterioration of health-related QoL during the induction period [57,58] while improvements were observed during the maintenance schedules [38,43,58].

Induction was shown to worsen cognitive functioning and insomnia with stable emotional functioning [57], physical functioning and fatigue with improved emotional functioning, urinary symptoms, future worries, and sexual functioning [58]. Regarding insomnia, 60–70% of patients rated their sleep quality poor resulting from decreased sleep efficacy with increases of mobile time, immobile bouts ≤ 1 min, and fragmentation index [57], which was consistent with the higher insomnia scores reported by the cross-sectional study conducted by Catto et al. [54]. The negative impact on sleep quality was believed to be caused by transient increase of voiding frequency, intermittency, and nocturia [57]. In a RCT study [48], induction treatment using half-dose of BCG resulted in better symptom QoL with almost all symptom scores decreased but a poorer function QoL when compared with the standard dose, which partially explained the stable global health and QoL states observed in other studies. However, in another RCT study, a 1/3 dose of BCG displayed negative impacts on general functioning and symptom QoL scales, and positive impacts on NMIBC-specific QoL scales but negative impacts on sexual functioning measured, when the measurements after the induction and end of maintenance treatment cycles were compared with the baseline values before the first instillation [44].

During and after the maintenance period, progressive improvements were observed in sexual activity, micturition problems, and activity levels [38] and different functioning dimensions and most of the symptoms [43] resulting in better global health status and overall QoL. In the RCT study conducted by Koga et al. [43], positive QoL impact was not observed in the observational group without receiving the maintenance treatment. Contradictorily, a single-arm clinical trial of BCG maintenance exhibited continuous deterioration in various functioning dimensions but improvement in a few non-specific symptom scores (nausea/vomiting, appetite loss, constipation) in addition to deteriorated insomnia and diarrhea, despite urinary symptoms and sexual function being improved together with a slightly improved global health status [58].

#### 3.3.3. Immunotherapy

Inclusive outcomes were reported in a small sampled RCT study was designed to compare the effects of intravesical IL-2 following incomplete TURBT with complete TURBT alone, although statistically insignificant positive and negative changes were observed in certain general- and NMIBC-specific QoL parameters in both arms [51].

#### 3.3.4. Chemotherapy

Among the 10 included studies involving chemotherapeutic agents, only 2 RCT [46,53] and 1 cohort [45] studies investigated the effects of chemotherapy on QoL in 3 different treatment protocols involving 2 drugs. Intravesical treatment schedule with MMC resulted in constant deteriorations in urinary symptoms and global health status throughout the 12-month follow-up period. Similarly, worsening of social function and global health status that were associated with local urinary symptoms in addition to increased fatigue, pain, appetite loss, and diarrhea were observed in the patient cohort receiving intravesical instillation with pirarubicin [45]. However, Huang et al. [53] demonstrated the positive impact of adding HA to the pirarubicin regimen on relieving the pelvic pain and urinary symptoms up to the 2-year follow-up period. The remaining studies involving chemotherapy that compared with BCG [36,44,47,50] and assisted the drug delivery using hyperthermia [42,52,56] will be discussed in the following two sections.

#### 3.3.5. Comparison between Intravesical BCG and Chemotherapy

Limited evidence suggested intravesical therapy with BCG or chemotherapeutic agents has its own pros and cons. A cohort study suggested that BCG was good in reducing pains over MMC while MMC performed better in problems with cystoscopy over BCG, despite both having a similar impact on QoL [50]. In a RCT study, GEM was shown to be superior to BCG for general QoL while BCG performed better in the bladder cancer-specific QoL than BCG, despite both exhibiting comparable negative impacts on sexual QoL [44]. When administered intravesically following TURBT, MMC demonstrated the improvement on sexual QoL which was contradictory to the negative impact of BCG, while BCG was shown to be beneficial to bowel symptoms over MMC, despite MMC resulting in a poorer BCI urinary summary score at 6 months but restored to score level equivalent to BCG [47]. Pilot results of a RCT study suggested the potential benefits of combining BCG with MMC on QoL related to local urinary symptoms [36].

#### 3.3.6. Hyperthermia

Post-trial measurement of a quadrimodal treatment that incorporated hyperthermia with cisplatin indicated that the vast majority of patients were satisfied with the QoL associated with urinary symptoms while only 3% of them rated themselves unhappy [42]. Patients treated with radiofrequency-induced thermo-chemotherapy with MMC were shown to have a slight increasing trend of health status from 3 to 9 months after instillation and rated higher health status than those treated with BCG, although statistically insignificantly [52]. Chemohyperthermia with MMC exhibited comparable effects with MMC instillation but superior effects over BCG on overall functioning QoL at the 4th week of induction treatment [56].

## 4. Discussion

The primary objective of this study was to review the existing scientific evidence on the impact of intravesical therapy towards QoL in NMIBC patients. To answer the PICOT question, the selected literatures comprised 6 studies allowing the comparison between intravesical therapy and TURBT, 14 studies having the focus on immunotherapy with 13 of them investigated BCG, and 10 studies focusing on chemotherapy with 4 of them were comparing with BCG with another 3 studies have adopted hyperthermia.

Adjuvant intravesical therapy following TURBT was not just a standard of care effective for preventing the recurrence of NMIBC [2], evidence collated in this current review supports its long-term positive impact on general and mental general QoL over TURBT alone. This is inconsistent with the findings of recent systematic reviews on bladder cancer surveillance that have suggested the negative impact on QoL due to the length and frequency of repeat intravesical therapy and cystoscopy [8,12]. There are two possible reasons for such inconsistency in the current review: first, because the actual effects were determined by comparing different time points to observe the long-term improvement of QoL; and second, the effects of follow-up and diagnosis in patients were ignored when determining the outcomes. Without a doubt, it is well known that the QOL of patients could be largely impacted by the protracted course of surveillance follow-up and care, whereas new diagnostic technologies such as the magnetic resonance imaging (MRI) without the need for contrast injection may be beneficial to improve the QoL [71]. In-depth analysis of the reviewed studies revealed similar positive QoL effects between BCG and certain chemotherapeutic drugs (for instance MMC, pirarubicin, and gemcitabine) over time, even though some symptom and functioning burdens were evidenced. The intravesical treatment protocol for high-risk NMIBC patients could last for more than 3 years [6]. During the induction period, BCG worsened the local urinary symptoms and sexual dysfunctions that resulted from TURBT, but continuous improvements were observed after beginning the maintenance period and remained stable afterwards. This is aligned with the early literature review published in 2003 [23]. In contrast, intravesical chemotherapy was more prone to being associated with non-specific symptoms such as pain and fatigue throughout the treatment period, although deteriorations in urinary symptoms and global health status could also occur. Interestingly, hyaluronic acid helped to ease the pain and urinary symptoms associated with pirarubicin for up to 2 years follow-up period, but no other QoL parameters were measured [46]. QoL data regarding chemotherapy remain limited [16]. On the other hand, hyperthermia has shown a positive trend in enhancing the QoL of patients receiving intravesical chemotherapy, which requires more investigations.

Owing to the reason of most interventions conducted on intravesical therapy being focused on evaluating effectiveness and toxicities without measuring QoL [13,14,15,16,19,22,61], the current systematic review collected more comprehensively a mixture of interventional and observational studies. In fact, the number of eligible studies included was still limited to cover existing intravesical modalities that are known to be effective. Regarding immunotherapy, aside from BCG, only a single study investigating the effects of interleukin-2 on QoL was identified which was insufficient to draw any conclusions [51], whilst many other agents such as interferon-alpha [19,20] and immune checkpoint inhibitors [72] were not addressed. Except for three identified studies on hyperthermia, other device-assisted modalities intravesical drug delivery such as EMDA [22,30] were completely lacking. Nothing has been updated from the systematic reviews on interferon-alpha [19] and EMDA [22] published 5 years ago, both reported no data related to disease-specific QoL was found. Moreover, the emerging photodynamic therapy has been shown to be highly effective and safe [32,33], whilst none of the studies have addressed the QoL in patients. No doubt, QoL outcomes of intravesical therapy were understudied in terms of both volume and coverage of different modalities. However, two-thirds of the eligible studies were published in the past 10 years which has projected an increasing trend of using QoL measures in recent interventions. With the identification of registered trials that have incorporated QoL outcome measures in their protocol while results were yet to be available, a more comprehensive outlook for QoL impacted by novel and effective intravesical therapies will be available in the 5–10 years. This aligns with the call for uncovering patient-report outcomes in bladder cancer [73].

Regarding the outcome measures, a total of 11 well-validated questionnaires or scales in addition to other self-developed instruments were among the 24 eligible studies for measuring different aspects of general and disease specific QoL. Since the literature review published by Bottemen et al. [23], the combination use of multiple QoL instruments for evaluating intravesical therapy has become a common practice that also resulted in generating heterogeneous outcome to hinder the effect size estimation to proceed for meta-analysis. The same phenomenon was concluded from another systematic review on NMIBC effectiveness trials that led to the recommendation of using the core set of QoL outcomes to reduce heterogeneity [24]. In addition to heterogeneity, another limitation of current review is the publication quality of the included studies. Quality assessment has determined that the majority of studies under review were poorly conducted or reported, which concurred with many other systematic reviews on effectiveness of intravesical therapies [16,17,19,22].

## 5. Conclusions

Existing evidence was insufficient to provide a conclusive answer to the PICOT question of this systematic review because data were not found to uncover the impact of many emerging intravesical therapies that have been suggested to be effective and safe. Limited studies revealed the long-term positive impact of intravesical immunotherapy with BCG or chemotherapy with certain drugs, irrespective of the worsening symptom burdens and functional performance observed during the induction treatment. However, treatment-specific QoL studies are required for newly emerging intravesical modalities that have displayed effectiveness.

## Figures and Tables

**Figure 1 ijerph-19-10825-f001:**
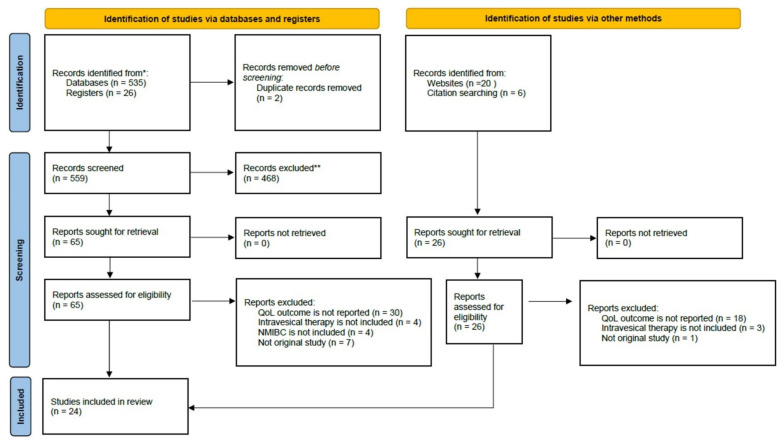
The process of literature screening and selection presented using the PRISMA 2020 flow diagram. * PubMed and the Cochrane library database; ** Studies did not involve NMIBC patients, did not include quality-of-life as outcome measures, and non-English publications.

**Figure 2 ijerph-19-10825-f002:**
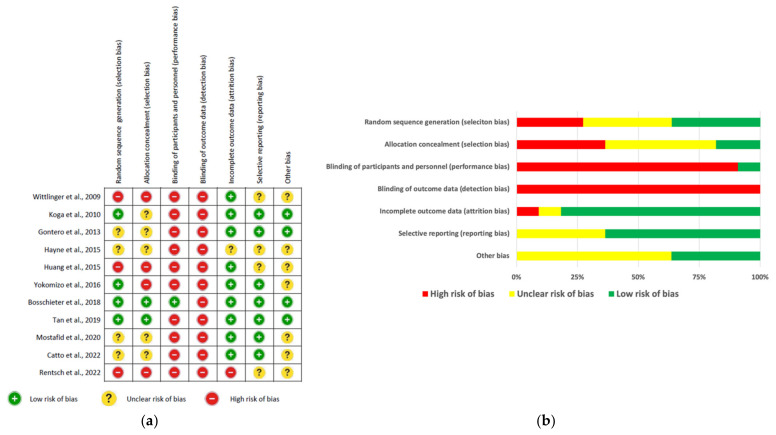
Summary of risk of bias results evaluated by using the CONSORT 2010 checklist for interventional studies: (**a**) The risk of bias levels of key RCT components rated in every included study; (**b**) the percentage of each risk of bias level of the key components [36,42,43,44,48,51,52,53,55,58].

**Figure 3 ijerph-19-10825-f003:**
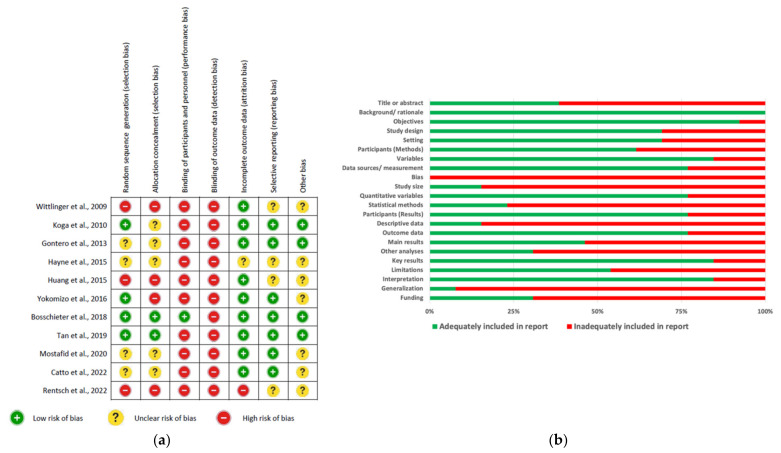
Summary of risk of bias results evaluated by using the STROBE Statement checklist for interventional studies: (**a**) The risk of bias levels of items rated in every included study; (**b**) the percentage of each risk of bias level of the items [36,42,43,44,48,51,52,53,54,58].

**Table 1 ijerph-19-10825-t001:** A summary of the characteristics and main findings of the eligible studies included in this systematic review.

Author, Year	Study Design	Cancer Stage, Grade Condition	Intravesical Agents Used	Comparison Groups (n)	QoL Instruments Variables	Time Points of Measurement	Main Findings
Bohle et al., 1996 [37]	Prospective cohort	Ta-T1 at grade 1–3	BCG	Single group (n = 30)	Satisfaction with life; health status (7-point scale)	2 weeks before (Post-TURBT), during (after 6th instillation), and 3-month after treatment	No statistically significant differences observed among the 3 time points for both QoL variables
Mack & Frick, 1996 [38]	Prospective cohort	Newly diagnosed (T1 or CIS at grade 3 (high-risk); Recurrent Ta-T1 at grade 1–2 (intermediate-risk)	BCG	Single group (n = 85)	Overall QoL; Sexual activity (3-point scale); Symptoms; Activity level (Distressed or not)	During induction cycle (T0), 1-month (T1) and 3-month (T2) during maintenance	84% and 23% rated bad-to-moderate levels of overall QoL and sexual activity at T0, but progressively improved during the maintenance. With 84% distressed with micturition problems at T0, and slightly improved during maintenance. Activity levels progressively improved as increased from 51% at T0 to 81% at T2.
Williams-Cox, 2004 [35]	Cross-sectional	Newly diagnosed and undergoing intravesical therapy	Not specified	Maintenance (n = 10) vs. Induction (n = 3)	Satisfaction on general QoL; Affecting daily life (5-point scale); QoL related to physical condition (7-point scale)	During induction or maintenance period	No significant difference between the two groups for QoL satisfaction ranged from ‘satisfied’ to ‘somewhat satisfied’ levels and physical condition at ‘good-to-very good’ levels. Patients rated ‘little or rather affected’ level of daily activity during induction treatment but improved to ‘not at all-to-little’ levels during maintenance.
Abbona, et al., 2007 [39]	Cross-sectional	Any stages received ≥1 intravesical cycle	Not specified	Single group (n = 63)	Working activity/relations; Sexual/couple life; Self-esteem (Number and percentage of concern)	Baseline data before entering psychosexual support therapy	Intravesical therapy altered working activity/relations, sexual/couple life, and self-esteem in 43%, 46%, and 38% of patients, respectively.
Gilbert et al., 2007 [40]	Cross-sectional	Ta-T2 or CIS	Not specified	Intravesical (n = 75) vs. No intravesical (n = 52)	BCI	Post-treatment	No significant difference between the groups in the urinary domain. Intravesical group exhibited better bowel function scores but lower function and higher bother scores for the sexual domain, despite statistically insignificant.
Sighinolfi et al., 2007 [41]	Prospective cohort	Ta-T1 or CIS at grade 2–3	BCG	Single group (n = 30)	IIEF-5; IPSS	During treatment, 1-month post-treatment	Erectile dysfunction and urinary symptoms significantly reduced after BCG treatment, in addition to a mean IIEF-5 score increased from 17.6 to 21.7 (*p* < 0.01) and a mean IPSS score decreased from 17.0 to 11.7 (*p* < 0.001). QoL levels due to urinary symptoms not reported.
Wittlinger et al., 2009 [42]	Single-arm trial	Primary or recurrent T1-2 or CIS	Hyperthermia with cisplatin	Single group (n = 30)	Item no. 8 of IPSS (IPSS-8)	Post-trial of Quadrimodal treatment	Vast majority of patients rated ≤3 for QoL due to urinary symptoms indicated satisfaction with only 3% of them rated unhappy (score 5).
Koga et al., 2010 [43]	RCT	Primary or recurrent Ta-T1 or CIS	BCG	Maintenance (n = 26) vs. No maintenance (n = 27)	EORTC QLQ-C30	Post-induction, 14 months	The maintenance group reported better QoL after 14 months, in terms of functioning (physical, role, social), global health status, and all symptom scores, except for nausea and vomiting and dyspnea, although statistically insignificant. Such improvements were not observed in the observational group without receiving maintenance.
Gontero et al., 2013 [44]	RCT	Ta-1 at grade 1–2	BCG; GEM	1/3 dose BCG (n = 47) vs. Gemcitabine (n = 41)	EORTC QLQ-C30; EORTC QLQ-BLS24	Pre-instillation (T0), after induction cycle (T1), and end of maintenance (T2)	When compared with GEM, BCG exhibited poorer QoL in all functioning scales at significance levels of *p* < 0.05 for cognitive and emotional at T1, *p* < 0.05 for physical and *p* < 0.01 for cognitive at T2. BCG exhibited better QoL for insomnia and appetite loss at T1, but poorer QoL in all symptoms at T2. However, BCG was corresponded with better performance than GEM in bladder cancer specific QoL in terms of urinary symptoms, intravesical treatment problem, and abdominal bloating and flatulence at both T1 and T2. Effects on sexual functioning are comparable negative effects between BCG and GEM at both T1 and T2.
Wei et al., 2014 [45]	Prospective Cohort	Intermediate- or high-risk NMIBC	Pirarubicin	Single group (n = 106)	EORTC QLQ-C30; CLSS	Pre- and post-instillation	Global health status and social functioning were worsened significantly (*p* < 0.0001) while fatigue, pain, appetite loss, diarrhea increased significantly (*p* < 0.01), after instillation. Following instillation, a significant reduction (*p* < 0.05) in CLSS QoL index from ‘pleased and mostly satisfied’ to ‘mixed and mostly dissatisfied’ levels.
Hayne et al., 2015 [36]	Pilot RCT	Resected, high-risk NMIBC	BCG; BCG + MMC	BCG (n = 11) vs. BCG + MMC (n = 12)	EORTC QLQ-C30 EORTC QLQ-BLS24 IPSS; ICSI	Baseline, 3, 6, 9, 12 months	Both groups followed the similar trend of changes in IPSS and cystitis scores throughout the follow-up period, except for an upward trend of cystitis scores exhibited by BCG from 9–12 months which was opposite to that by the BCG + MMC group. BCG + MMC showed slightly better (lower) QoL scores at all times. Results of EORTC questionnaires were not reported.
Huang et al., 2015 [46]	RCT	Ta-T1 at grade 1–3	Pirarubicin + HA	Pirarubicin + HA (n = 64) vs. Pirarubicin + Placebo (n = 63)	Pain VAS (10 cm)	Monthly since baseline up to 2 years	Addition of HA improved the pain consistently after 1 month of treatment (*p* < 0.05) and beyond, as associated with the perceptible relief of pelvic pain and urinary symptoms.
Schmidt et al., 2015 [47]	Prospective cohort	Ta-T1 or CIS at grade 1–3	BCG MMC	TUR (n = 144) vs. TUR + BCG (n = 51) vs. TUR + MMC (n = 31)	SF-36; BCI	6 months (T1) and 12 months (T2) post-diagnosis	Only reported change in scores between time points. All three treatments demonstrated negative impacts on SF-36 physical health component summary, whereas intravesical treatments showed to be beneficial on mental health, with particular better scores in both T1 and T2 in the MMC group but only at T2 in the BCG group. MMC resulted in poorer BCI urinary summary score at T1 but restored to score level equivalent to BCG and TURBT at T2. BCG showed to be beneficial to BCI bowel summary scores in both time points. Whilst results of TUR + MMC (improvement) on sexual summary scores were contradictory to TURBT + BCG and TURBT alone, both showed negative impacts.
Yokomizo et al., 2015 [48]	RCT	Ta-T1 or CIS at grade 1–3	BCG	Standard dose (n = 79) vs. Half-dose (n = 79)	EORTC QLQ-C30	Baseline and after induction	The half-dose was corresponded with poorer functioning but better symptom QoL. When compared with the baseline, after induction, standard dose resulted in improved global QoL and functioning scores, except emotion function. Half-dose decreased all symptom scores, but standard dose increased diarrhea and constipation scores.
Danielsson 2018 [49]	Prospective cohort	T1 at grade 2–3	BCG	Single group (n = 113)	Urinary bladder symptoms and symptom burdens (Self-developed 17-item questionnaire	Baseline (before 1st dose), during instillation (at 3rd, 6th, 12th, and 18th), after treatment (at 24th and 60th month)	Progress improvement by BCG observed over 12 months in prevalence, intensity, and burden caused by several urinary bladder symptoms that have been reported by patients prior to the treatment. General health remained stable over time. Percentage of patients rated ‘moderate large’ total symptom burden progressed reduced from 31% at baseline to 15% over the 24 months and further reduced to 5% at 60 months. Percentage of patients rated ‘Bad/very bad’ mental well-being reduced from 14% at baseline to 12% at 3 months, then to 5% at 6 months and remained stable over time.
Siracusano et al., 2018 [50]	Prospective cohort	Intermediate- or high-risk NMIBC	BCG; MMC	Single (n = 108)	EORTC QLQ-C30; EORTC QLQ-BLS24	Baseline (T0), 6th or 8th week after 1st instillation (T1), and 3 months after induction (T2)	Global QoL, functioning QoL except cognitive, and cancer-specific symptoms (Pain, dyspnea, insomnia) significantly deteriorated at T1 but improved at T2, as measured by QLQ-C30. Patients encountered significant (*p* < 0.05) deterioration in urinary symptoms, worry about future disease, erection problem (male), and feeling uneasy about sexual intercourse at T1, whereas these conditions were improved at T2 and they were even significantly (*p* < 0.05) better than T0. Scores for problem about cystoscopy exam and abdominal bloating flatulence were also significantly improved (*p* < 0.001). Stratified analysis indicated that BCG and MMC were similar in their impacts on QoL. BCG is good at reducing pains over MMC while MMC performed better in problem about cystoscopy exam over BCG
Bosschieter et al., 2019 [51]	RCT	Ta at grade 1–2	IL-2	Complete TURBT (n = 14) vs. Incomplete TURBT + BCG (n = 14)	EORTC QLQ-BLS24; SF-36	Baseline, 3 months after TURBT (Outcomes reported as mean difference changed from baseline)	No statistically differences between the two arms when comparing the mean score differences. Compared with the baseline, addition of IL-2 following incomplete TURBT impacted negatively on physical and social functioning, bodily pain, vitality (SF-36) and most of the urinary symptoms (BLS-24) but positively on physical functioning, mental health, and general health perception (SF-36) and erectile dysfunction (BLS-24) at 3 months.
Tan et al., 2019 [52]	RCT	Recurrent, Ta-T1 or CIS at grade 1–3	BCG; RITA-MMC	BCG (n = 56) vs. RITA-MMC (n = 48)	EQ-5D	Baseline, at 3rd, 6th, 9th, 12th month of treatment	No statistical difference was observed between BCG and RITA-MMC arms for health status, although higher EQ-5D index scores were shown at 3th, 6th, 9th months.
Mostafid et al., 2020 [53]	RCT	Ta-T1 or CIS at grade 1–2; EORTC risk of recurrence score ≤ 6	MMC	MMC (n = 54) vs. Surgery (n = 28)	EORTC QLQ-C30; EORTC QLQ-BLS24	Baseline, 3rd, 6th, 12th month (Outcomes reported as mean difference changed from baseline)	For global health status and urinary symptoms, surgery had no impact in the first 6 months but deteriorated at the 12th month. Deteriorations observed in MMC throughout the follow-up period. Both arms did not improve physical function but deteriorated by surgery at the 12th month.
Catto et al., 2021 [54]	Cross-sectional	Ta-T1 or CIS	Not specified	TURBT + BCG/MMC (n = 562) vs. TURBT alone (n = 306)	EQ-5D; EORTC QLQ-BLM30; QLQ-NMIBC24	1–10 years after treatment	Intravesical therapy following TURBT caused higher percentage of pain/discomfort but lower percentage of anxiety/depression than TURBT alone as measured by ED-5D. Patients received intravesical therapy also reported mean score of 5.6 for specific treatment issues with slightly higher global QoL and functioning scores except for cognitive but lower sexual function and higher male sexual problems scores, in addition to slightly lower symptom scores (fatigue, nausea and vomiting, dyspnea, appetite loss, and diarrhea) but higher score for insomnia than TURBT alone.
Catto et al., 2021 [55]	RCT	High-grade Ta-1 or CIS	BCG	BCG (n = 25) vs. RC (n = 25)	EQ-5D; EORTC QLQ-C30; EORTC QLQ-NMIBC24	Baseline; 3, 6, and 12 months follow-up	No differences were observed between BCG and RC in the trends of EQ5D QoL and QLQ-C30 functioning QoL scores, except RC showed reductions on global health status, physical and role functioning score at 3 months, and social functioning score at 6 months. For the treatment-specific QoL measured by NIMIBC24, BCG increased slightly the urinary symptom scores at 6 months and concerns about contaminating a partner at 3 months, which restored to baseline levels between 6 and 12 months. Small reductions in future worry scores were also observed with time. As compared with RC, BCG group reported higher urinary symptom scores between 6 and 12 months, lower sexual function score at 6 months, lower future worries and bloating and flatulence scores at 3 months.
Gonzalez-Padilla et al., 2021 [56]	Prospective cohort	Intermediate-risk and high-risk of Ta-1 and CIS	BCG; MMC; CHT-MMC	BCG (n = 27) vs. CHT-MMC (n = 14) vs. MMT (n = 15)	FACT-BI; IPSS	Baseline, at 4th and 6th week, and 1 week after induction	All 3 study groups followed similar trends in improving the symptom scores measured by FACT-BI and IPSS. CHT-MMC showed superior effects over BCG but that was comparable to MMC alone. Statistical significance (*p* < 0.05) observed between CHT-MMC and BCG for FACT-BI scores measured at 4th week of induction. Among all 3 groups, no significant change was observed in the QoL due to LUTS as measured by the IPSS item 8.
Miyake et al., 2022 [57]	Prospective cohort	Intermediate or high-risk NMIBC; EORTC performance status 0–1	BCG	Single group (n = 10)	IPSS; EORTC QLQ-C30; FACT-BI; SF-8; Sleep quality (MotionWatch8)	Baseline; 4th and 8th induction doses; 1 month after induction treatment	Significant deteriorated changes observed in cognitive functioning, insomnia, during and at 1 month after induction. The sleep quality also significantly deteriorated (in terms of decreased sleep efficacy, increased mobile time, increased immobile bouts ≤ 1 min, and increased fragmentation index with 60–70% rated poor sleep quality) during induction period, but restored to the baseline levels after 1 month. Stable emotional functioning and symptom (pain, dyspnea) scores observed during induction, but rebound to pre-TURBT levels after 1 month. Constipation scores increased at post-TURBT and remained stable during and at 1 month after induction. Appetite loss reduced at post-TURBT, but increased during induction and rebound to pre-TURBT level after 1 month. Transient increase in urinary symptom (frequency, intermittency, nocturia) and IPSS total scores during induction that caused negative impact on sleep quality.
Rentsh et al., 2022 [58]	Single-arm trial	Recurrent NMIBC at intermediate or high-risk for progression (EORTC score 7–23); post BCG	BCG	Single group (n = 40)	EORTC QLQ-C30; EORTC QLQ-NMIBC24	Before induction (T0), before maintenance (T1), and end of treatment (T2)	Induction therapy exhibited mild effects on QLQ-C30 domains, but 49% of patients reported improvement in emotional functioning while one-third of them reported deterioration in global health status (33%), physical functioning (30%), and fatigue (30%). For QLQ-NMIBC24 measures, induction therapy deteriorated urinary symptoms, future worries, and sexual function especially for males while it improved intravesical treatment issues. Maintenance treatment improved global health status slightly but deteriorated role and social functioning, in addition to physical and cognitive functioning that has already worsened during induction. The improvement in emotional functioning persisted after treatment. However, it improved a few symptom scores (nausea/vomiting, appetite loss, constipation) but deteriorated insomnia and diarrhea. Urinary symptoms and sexual function improved while sexual problems in males remain unchanged. Intravesical treatment issues deteriorated after maintenance therapy.

BCI = Bladder Cancer Index; CHT-MMC = Chemohyperthermia with MMC; CLSS = Core Lower Urinary Tract Symptom Score; EORTC QLQ-C30 = European Organization of Research and Treatment of Cancer Quality of Life Questionnaire; EORTC QLQ-BLS24/NMIBC24 = Non-muscle invasive bladder cancer module of the EORTC-QLQ; EQ-5D: EuroQoL-5 dimensions; FACT-BI = Bladder cancer-specific symptoms; GEM = Gemcitabine; HA = Hyaluronic acid; ICSI = Interstitial Cystitis Symptom Index; IIEF-5 = The International Index of Erectile function; IPSS: International Prostate Symptom Score; MMC = Mitomycin C; RITA = Radiofrequency-induced thermos-chemotherapy; SF-36 = Short Form Health Survey Questionnaire; SF-8 = 8 items version of SF-36.

## Data Availability

All data generated or analyzed during this study are included in this systematic review article.

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
