# Peer review of "Impact of Effective Intravesical Therapies on Quality of Life in Patients with Non-Muscle Invasive Bladder Cancer: A Systematic Review"

_ijerph, 2022, doi:10.3390/ijerph191710825_

Round 1
Reviewer 1 Report
This article proposes an unrevealed issue of the impact of intravesical modalities on patients’ quality of life. These intravesical modalities are indeed more effective for NMIBC than systemic chemotherapy, whereas the recent evidence of the QoL of patients after receiving intravesical modalities remains inconsistent. Authors exhaustively collated the current QoL-related evidence and pointed out this problematic issue of the field that may be attractive for the readership.
1. Some issues should be further described or clarified.
For the broader readership, authors could add the advantage of intravesical therapies, the characteristic of bladder urothelium that facilitates the practice of intravesical therapies, and the applications of intravesical drug delivery to bladder disorders other than bladder cancer.
Authors could add the utilisation rates of intravesical modalities in NMIBC.
2. Some problems should be amended, which are as follows:
Consort 2010 checklist should be “CONSORT” 2010 checklist.
Strobe statement checklist should be “STROBE” statement checklist.
In line 25, “lacking for” is wrong grammar.
In line 28, “uncover” may not be the optimal description. Perhaps “clarify”.
In line 29, “demands for” is wrong, just “demands”.
In line 54, lacking data “of”.
Line 60. “while they were improved…” should be “,these symptoms were improve…”
Line 67. “new modalities emerged to demonstrate at least comparable if not better prophylactic effectiveness but less toxicities” is confused. Please check the grammar.
In lines 92-94. Is “therap” a typo?
In line 144. Is {Williams-Cox, 2004} the reference?
In line 395. and 10 studies “focusing” on chemotherapy
Author Response
As attached.

Reviewer 2 Report
This is a well written manuscript about QoL in patients with bladder cancer.
I have no major comment. I would suggest to the authors to include in the discussion a comment also about follow-up and diagnosis in these patients, that could seriously affect the QoL. The importance of new technologies, could be an important aspect, as for example MRI (please see the manuscript by Delli Pizzi et al. 10.1007/s00330-020-07473-6). Authors should consider to discuss the beneficial effect of these technologies.
Author Response
As attached.

Reviewer 3 Report
Please see the attachment

Author Response
As attached.
